# Lightweight End-to-end Text-to-speech Synthesis for low resource on-device applications

*Biel Tura Vecino, Adam Gabryś, Daniel Mątwicki, Andrzej Pomirski, Tom Iddon,*
*Marius Cotescu, Jaime Lorenzo-Trueba*

Alexa AI

bieltura@amazon.com

## Abstract

Recent works have shown that modelling raw waveform directly from text in an end-to-end (E2E) fashion produces more natural-sounding speech than traditional neural text-to-speech (TTS) systems based on a cascade or two-stage approach. However, current E2E state-of-the-art models are computationally complex and memory-consuming, making them unsuitable for real-time offline on-device applications in low-resource scenarios. To address this issue, we propose a Lightweight E2E-TTS (LE2E) model that generates high-quality speech requiring minimal computational resources. We evaluate the proposed model on the LJSpeech dataset and show that it achieves state-of-the-art performance while being up to 90% smaller in terms of model parameters and $10\times$ faster in real-time-factor. Furthermore, we demonstrate that the proposed E2E training paradigm achieves better quality compared to an equivalent architecture trained in a two-stage approach. Our results suggest that LE2E is a promising approach for developing real-time, high quality, low-resource TTS applications for on-device applications.

**Index Terms**: speech synthesis, text-to-speech, end-to-end, on-device.

## 1. Introduction

Text-to-speech (TTS) technology has come a long way in recent years, with state-of-the-art (SOTA) models generating highly realistic and natural-sounding speech [1]. Given its success, TTS technology is now widely used in many different applications, either with cloud-based online connections or offline synthesis. However, as the demand for on-device, real-time speech synthesis grows, offline TTS systems are becoming increasingly important in today's society. With the rise of smart devices and the Internet of Things (IoT), there is a need for TTS systems that can function without the need for a remote server connection, providing users with instant and reliable access to speech synthesis. Offline on-device TTS systems are particularly useful in scenarios where internet access is limited, unreliable, or where privacy concerns are critical to the application. Hence, the development of offline TTS systems is essential to fulfill the requirements of diverse applications.

Text-to-speech (TTS) models are typically composed of two components: an acoustic model that predicts acoustic units from the input text, and a vocoder model that generates the speech waveform from these acoustic features. However, the two-stage approach of training separate models leads to a mismatch between the acoustic features used during training and those used during inference [2], resulting in a degradation of synthesis quality. This occurs because the predicted acoustic features by the acoustic model, usually in the form of a mel-spectrogram [3, 4], may not precisely match the ones used in

the training process of the vocoder. To overcome this issue, a fine-tune process is performed to a pre-trained vocoder model on predicted acoustic features [5, 6]. This approach requires different sets of hyperparameters and training/fine-tuning procedures for each model, which can lead to suboptimal performance and a more complex pipeline.

In this paper, we propose a new end-to-end text-to-speech (E2E-TTS) model based on a joint training of a lightweight acoustic and vocoder model in a single efficient architecture, enabling the benefits of end-to-end speech modeling to be applied to on-device TTS systems. Our proposed model is significantly smaller than other E2E solutions while maintaining comparable performance, making our approach more practical and accessible for low-resource scenarios. The main contributions of our work are as follows: 1) we present the Lightweight E2E-TTS (LE2E) model, based on a joint training paradigm of LightSpeech [7] and Multi-Band MelGAN [8] which outperforms its originally designed two-step cascade approach while only requiring a single joint training scheme; 2) we introduce an upgraded loss objective based on recent GAN speech discriminators, which we show to be effective not only on a known neural vocoder architecture but also in an E2E-TTS system and; 3) we show that the proposed model achieves a mean opinion score (MOS) of 3.79 in the LJSpeech dataset, on par with VITS [9] and just below JETS [10], while being much more memory-efficient and faster, suitable for offline on-device applications.

## 2. Related work

Several recent studies have proposed removing the mel-spectrogram as an intermediate representation for E2E-TTS synthesis [11, 9]. FastSpeech2 [12] uses Parallel WaveGAN [13] to synthesize speech directly from text, while VITS [9] and NaturalSpeech [14] proposes a flow-based TTS system trained jointly with the HiFi-GAN [5] vocoder to generate waveforms from text sequences. JETS [10] improves upon FastSpeech2 by jointly training with the HiFi-GAN vocoder. Both JETS and VITS have shown impressive results, but they are not suitable for on-device low resource real-time synthesis due to their high computational requirements. In contrast, low-memory footprint TTS systems like LightSpeech [7], SpeedySpeech [15] and FCH-TTS [16] strive to minimize computational power, but operate on a cascade approach and require specific vocoder architectures. LiteTTS [17] is a lightweight mel-spectrogram-free TTS system that achieved competitive results compared to state-of-the-art E2E models, but employs the memory-heavy HiFi-GAN as vocoder. Low-resource E2E speech synthesis models have been explored in works like [18] which propose an autoregressive model or in [19] which present a lightweight flow-based E2E architecture for on-device applications.

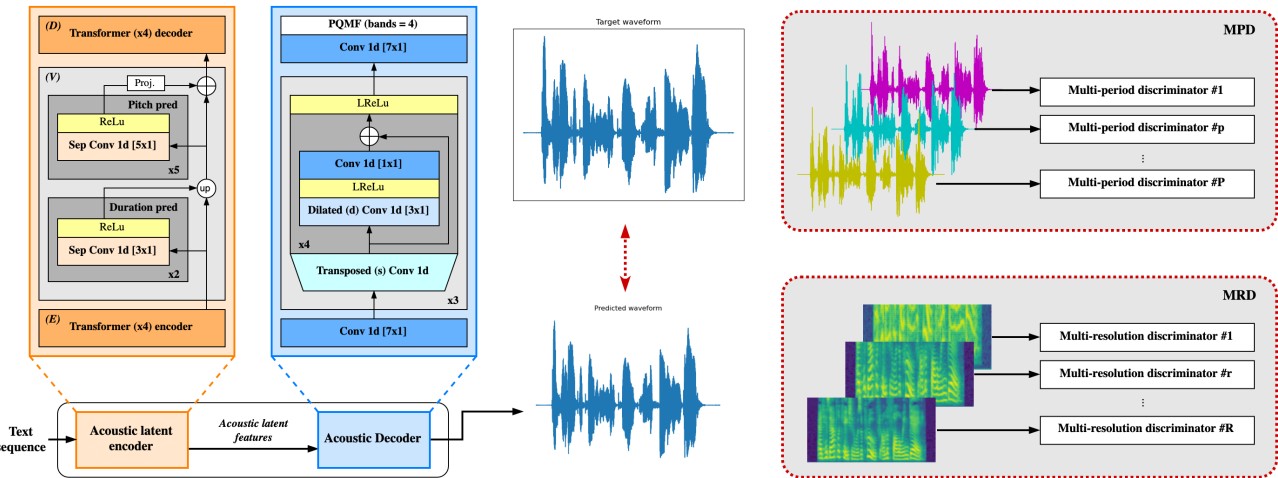

Figure 1: *LE2E model architecture and training discriminators.*

# 3. Methodology

The proposed model is shown in Figure 1. LE2E follows the typical GAN-based setup of a generator (G) that includes an acoustic latent encoder and an acoustic decoder, and a set of discriminators (D) divided in two different sets of multi-period (MPD) and multi-resolution discriminators (MRD).

## 3.1. Generator

The waveform generator consists of two components: an acoustic latent model and a neural vocoder, which are concatenated into a jointly trained single architecture. The acoustic latent model is inspired by the LightSpeech [7] model, but instead of predicting mel-spectrograms, it is trained to produce unsupervised acoustic latents that are used as input by the vocoder. The model takes phoneme and positional embeddings and outputs latent frame-level up-sampled acoustic embeddings. It is divided in three components: a text encoder (E), a variance adaptor (V) and an acoustic decoder (D). The text encoder generates positional aware phoneme embeddings through a stack of transformer layers. Then, these are fed into the variance adaptor. The variance adaptor is comprised of a duration predictor and a pitch predictor. The duration predictor takes output of the text encoder and predicts phoneme-level durations, used to up-sample the phoneme embeddings. Then, the up-sampled phoneme embeddings are fed to the pitch predictor, which is trained to predict frame-level pitch latents that are added up to the up-sampled phoneme embeddings. Finally, the output of the variance adaptor is inputted to the acoustic decoder. The acoustic decoder follows the exact same stack of transformer layer as the phoneme encoder. The vocoder model generates waveform signals from the latent embeddings produced by the acoustic model. The model is based on Multi-Band MelGAN [8] and takes the intermediate acoustic latents and upsamples them to generate the waveform. It is composed of a stack of up-sampling blocks that follow a transposed convolution and a set of residual blocks with dilated convolutions that increase the receptive field. The proposed vocoder architecture follows the method in [20] and generates different waveform sub-bands that are then combined to generate the final output waveform through Pseudo Quadrature Mirror Filter bank (PQMF).

## 3.2. Discriminators

State-of-the-art E2E-TTS systems use multiple discriminators that guide the generator in synthesizing a coherent waveform while minimizing perceptual artifacts that can be easily distinguished by a human ear. We adopted the set of discriminators proposed in BigVGAN [21], which includes a multi-period discriminator (MPD) and a multi-resolution discriminator (MRD). It is important to note that each discriminator comprises several sub-discriminators that operate on different resolution windows of the waveform. The MPD reshapes the predicted waveform into 2D representations with varying heights and widths to separately capture multiple periodic structures. On the other hand, the MRD is composed of several sub-discriminators that operate on multiple linear spectrograms with different short-time Fourier transform (STFT) resolutions.

## 3.3. Training objectives

The training objective of LE2E includes losses applied to the duration predictor, pitch predictor and waveform level, which include GAN and regression losses in form of power loss and multi-resolution STFT loss.

### 3.3.1. Duration loss

The acoustic latent model takes the hidden representation of the input phonemes and predicts the frame-level duration of each one of them in the logarithmic scale. The duration predictor is optimized with mean square error (MSE) loss, $\mathcal{L}_{dur}$ between predicted and oracle durations extracted by an external aligner based on the Kaldi Speech Recognition Toolkit [22] used in our experiments.

### 3.3.2. Pitch loss

Pitch prediction is typically achieved through regression tasks that estimate the exact pitch value [12, 23]. However, due to the high variability of ground-truth pitch contours, we replace the regression task with cross-entropy density modeling. To accomplish this, we follow [24] and apply a 256-bin quantization to the standardized pitch signal, followed by a cross-entropy loss function, $\mathcal{L}_{f0}$ on top of softmaxed predicted pitch logits.

### 3.3.3. Adversarial training

For simplicity with the notation, we name all the $K$ discriminators from MPD and MRD under the set of discriminators $D_k$. **GAN Loss**. The main adversarial training objective follows the least squares loss functions for non-vanishing gradient flows. The discriminator is trained to classify target samples to 1 and predicted samples to 0. The generator is trained to fake the discriminator by updating the sample quality to be classified.

$$\mathcal{L}_D(\boldsymbol{x}, \hat{\boldsymbol{x}}) = \min_{D_k} \mathbb{E}_{\boldsymbol{x}} \left[ (D_k(\boldsymbol{x} - 1)^2 \right] + \mathbb{E}_{\hat{\boldsymbol{x}}} \left[ (D_k(\hat{\boldsymbol{x}})^2 \right] \quad (1)$$

$$\mathcal{L}_G(\hat{\boldsymbol{x}}) = \min_{G} \mathbb{E}_{\hat{\boldsymbol{x}}} \left[ (D_k(\hat{\boldsymbol{x}} - 1)^2 \right] \quad (2)$$

**Feature matching loss**. Used in [25], it is defined as the similarity metric measured by the difference in features of discriminators between a ground truth sample and a generated sample. Feature matching loss is computed as the L1 distance between target and predicted discriminator hidden intermediate features and it is defined as:

$$\mathcal{L}_{FM}(\boldsymbol{x}, \hat{\boldsymbol{x}}) = \mathbb{E}_{\boldsymbol{x}, \hat{\boldsymbol{x}}} \left[ \sum_{i=1}^{T} \frac{1}{N_i} \left\| D_k^{(i)}(\boldsymbol{x}) - D_k^{(i)}(\hat{\boldsymbol{x}}) \right\|_1 \right] \quad (3)$$

### 3.3.4. Reconstruction losses

Applying a reconstruction loss to GAN models helps to generate realistic results [13]. We utilize two widely used reconstruction losses as auxiliary objectives to the GAN-based training. **Multi-resolution STFT loss**. This loss is defined as the sum of spectral convergence $\mathcal{L}_{sc}$ and STFT magnitude $\mathcal{L}_{mag}$ between predicted $\hat{s}$ and target $s$ STFT linear spectrograms:

$$\mathcal{L}_{sc}(\boldsymbol{s}, \hat{\boldsymbol{s}}) = \frac{\|\boldsymbol{s} - \hat{\boldsymbol{s}}\|}{\|\boldsymbol{s}\|_F}, \quad \mathcal{L}_{mag}(\boldsymbol{s}, \hat{\boldsymbol{s}}) = \frac{1}{S} \|\log(\boldsymbol{s}) - \log(\hat{\boldsymbol{s}})\| \quad (4)$$

$$\mathcal{L}_{STFT}(\boldsymbol{x}, \hat{\boldsymbol{x}}) = \frac{1}{M} \sum_{m=1}^{M} \mathbb{E}_{\boldsymbol{x}, \hat{\boldsymbol{x}}} \left[ \mathcal{L}_{sc}(\boldsymbol{s}, \hat{\boldsymbol{s}}) + \mathcal{L}_{mag}(\boldsymbol{s}, \hat{\boldsymbol{s}}) \right] \quad (5)$$

where $\|\cdot\|_F$ denotes the Frobenius norm and $\|\cdot\|_1$ the L1 norm, $S$ refers to the total number of values in both time and channel dimension in the linear spectrogram and $M$ denotes the number of different STFT resolutions, which coincide with the different inputs of the MRD. Note that following [8] we apply the defined multi-resolution STFT loss in both *full*-band and *sub*-band predictions. Therefore, the final multi-resolution STFT used is:

$$\mathcal{L}_{STFT}(\boldsymbol{x}, \hat{\boldsymbol{x}}) = \frac{1}{2} \left( \mathcal{L}_{STFT}^{full}(\boldsymbol{x}, \hat{\boldsymbol{x}}) + \mathcal{L}_{STFT}^{sub}(\boldsymbol{x}, \hat{\boldsymbol{x}}) \right) \quad (6)$$

**Mel-Spectrogram loss**. In addition to the multi-resolution STFT loss, we also incorporate a mel-spectrogram loss, also known as power loss, in the full-band prediction to improve the training stability [21, 5] . It is defined as the L1 norm between the predicted $\hat{\boldsymbol{m}}$ and target $\boldsymbol{m}$ mel-spectrogram extracted with the same parameters as in [5]:

$$\mathcal{L}_{mel}(\boldsymbol{x}, \hat{\boldsymbol{x}}) = \mathbb{E}_{\boldsymbol{x}, \hat{\boldsymbol{x}}} \left[ \|\boldsymbol{m} - \hat{\boldsymbol{m}}\|_1 \right] \quad (7)$$

### 3.3.5. Total loss

Summing up all the described loss functions, we end up with the final loss $\mathcal{L}$ for the generator in the jointly E2E training of the proposed architecture:

$$\mathcal{L} = \mathcal{L}_{dur} + \mathcal{L}_{f0} + \mathcal{L}_G + \lambda_{FM}\mathcal{L}_{FM} + \\ + \lambda_{mel}\mathcal{L}_{mel} + \lambda_{STFT}\mathcal{L}_{STFT} \quad (8)$$

where we set $\lambda_{FM} = 2$, $\lambda_{mel} = 5$ and $\lambda_{STFT} = 2.5$. The architecture is optimized to minimize the total loss $\mathcal{L}$ in addition with the discriminator loss $\mathcal{L}_D$ in an adversarial training approach.

## 4. Experiments and results

### 4.1. Experimental setup

For reporting the results of the proposed model and for easy comparison of our architecture, we evaluate our model on the widely used LJSpeech dataset [26]. LJSpeech consists of 13.100 pairs of text and speech data with approximately 24 hours of speech. We split the dataset into three parts: 12.900 samples for training and 100 samples for both validation and test set.

### 4.2. Model details

**Generator** The generator of LE2E is built upon two main components: the acoustic latent model and the vocoder model. The acoustic model follows a 4 block transformer phoneme encoder with self-attention. The dimension of the phoneme embeddings and hidden sizes of the self-attention are 256. The kernel size of the separable convolutional layers within each transformer layer follow $[5, 25, 13, 9]$ respectively. The duration predictor is a 2-layer 1D separable convolutional neural network with kernel-size 3. The pitch predictor is a 5-layer 1D separable convolution neural network with kernel-size 5 followed by a linear projection layer to 256 hidden dimensionality for pitch logits. The decoder follows the same architecture as the encoder, where each separable convolution has its kernel size to $[17, 21, 9, 13]$ respectively. As for the vocoder module, $300\times$ upsampling is conducted through 3 upsampling layers with $[3, 5, 5]$ upsampling factors respectively and a PQMF synthesis filter. The output channels of each upsampling layer are $[192, 96, 48]$ and each transposed convolution in the upsampling layer has its kernel-size of $[6, 10, 10]$ respectively. Each upsampling layer has 4 stacked residual blocks consisting of a 1D dilated convolution with kernel-size 3, a Leaky Relu activation with 0.2 slope, and a final 1D convolution with kernel-size 1. The dilation component in each dilated convolution of the 4 residual blocks follow $[1, 3, 9, 27]$ respectively. The final 1D convolution has a kernel-size of 7 and 4 output channels, which they get combined through a carefully designed PQMF filter with 62 taps, $\beta = 0.9$ and a cutoff ratio of 0.1492.

**Discriminator** LE2E discriminators are divided into the MRD and MPD module. Each module contains a set of sub-discriminators that use a stack of $2D$ convolutions followed by ReLU activations. In the MPD, the input waveform is first reshaped into a 2D signal by concatenating samples every $[2, 3, 5, 7, 11]$ samples (period) with reflective padding. In the MRD, the input is a linear spectrogram with a variable number of fast Fourier transform (FFT) points: $[1024, 2048, 512]$ with respective hop lengths of $[120, 240, 50]$ and window lengths of $[600, 1200, 240]$.

Table 1: *Evaluation metrics results on the LJSpeech dataset validation split. MOS is reported with a 95% confidence interval (CI)*

| Model | cFSD ($\downarrow$) | F0 RMSE ($\downarrow$) | XWLM ($\uparrow$) | MOS (CI) ($\uparrow$) |
|---|---|---|---|---|
| Recordings | - | - | - | 4.25 ($\pm$0.10) |
| VITS [9] | 0.254 | $0.042 \pm 0.024$ | $0.985 \pm 0.006$ | 3.81 ($\pm$0.14) |
| FastSpeech2 + HiFi-GAN (JETS [10]) | 0.212 | $0.041 \pm 0.058$ | $0.979 \pm 0.011$ | 4.01 ($\pm$0.13) |
| LightSpeech + MB-MelGAN+ (cascade) | 0.248 | $0.029 \pm 0.028$ | $0.968 \pm 0.016$ | 3.73 ($\pm$0.14) |
| LightSpeech + MB-MelGAN+ (LE2E) | 0.167 | $0.033 \pm 0.027$ | $0.972 \pm 0.017$ | 3.79 ($\pm$0.14) |

**Training process** The model was trained for 1M steps with an effective batch size of 128 samples. We used the AdamW [27] optimizer with $\beta_1 = 0.8$ and $\beta_2 = 0.99$ and an initial learning rate of $2 \times 10^{-4}$. We used an exponential decay scheduler that reduced the learning rate by a factor $\gamma = 0.99$ at each training epoch and a weight decay penalty factor of $1e-2$.

### 4.3. Evaluation metrics

To measure and compare the quality of the proposed model, we used a combination of three objective metrics and a subjective mean opinion score (MOS) evaluation. For objective evaluation, first we evaluated signal quality using the conditional Fréchet Speech Distance (cFSD). To compute cFSD, we generated activation distributions for both the recordings and the synthesized samples using a pre-trained XLSR-53 [28] wav2vec-2.0 [29] model. We then compared the distributions using the Fréchet Distance metric. Second, to assess the intonation fidelity, we computed the root mean-squared error (RMSE) of the fundamental frequency (F0) between the predicted waveforms and the recordings. A lower RMSE indicates higher F0 fidelity. Third, we evaluated the similarity between the recordings and the generated speaker embeddings using the mean cosine distance metric between extracted embeddings obtained through the pre-trained XVector head from WavLM [30] (XWLM). For the MOS evaluation, we gathered 100 US English speakers from the crowd-sourcing platform ClickWorker to evaluate the audio quality of 20 random samples from the test set. Each speaker evaluated 8 test cases and rated each sample on a scale of 1 (very low quality) to 5 (very high quality). We collected a total of 40 ratings for each sample to assess the subjective quality of the synthesized speech.

### 4.4. Results

**Vocoder ablation study** In order to show that the new proposed loss objective has a positive impact in our standalone neural vocoder architecture, we trained the Multi-Band MelGAN model with the original loss objective [8] and with the proposed loss described in Section 2.3. We will reefer to this latter as Multi-Band MelGAN+. Both architectures were trained on the same LJSpeech dataset and evaluated in the re-synthesis task of generating waveform signals from the test split. Following the original training paradigm, we pre-trained the generator for 200K steps and then train the whole architecture for 1M steps in both models. We used Adam [31] optimizer with a learning rate of $1e-4$ and a batch size of 128. Subjective MOS in Table 2 clearly shows that the proposed loss objective generates better quality waveform compared to original training objective without any change in the model architecture.

**Model comparison**. We compared the proposed LE2E architecture against two state-of-the-art E2E models: VITS [9] and JETS [10]. Both models are obtained from the *ESPNet* [32]

Table 2: *MOS comparison between recordings and the same vocoder model (MB-MelGAN) with different training paradigm.*

| Model | MOS (CI) ($\uparrow$) |
|---|---|
| Recordings | 4.24 ($\pm$0.13) |
| MB-MelGAN+ | 4.02 ($\pm$0.14) |
| MB-MelGAN [8] | 3.59 ($\pm$0.14) |

open-source implementation, in which a checkpoint of each model trained on LJSpeech dataset [26] is available. In addition, we trained the LightSpeech model to predict mel-spectrograms following the original implementation [7] to demonstrate that the proposed training paradigm improves the traditional cascade approach. To do so, we generated predicted mel-spectrograms from it and fine-tuned the proposed MB-MelGAN+ vocoder for an extra 200K steps to mitigate the domain mismatch in text-to-speech inference. Table 1 summarizes the comparison results, while Table 3 presents the memory consumption and computational complexity. LE2E does not only perform slightly better than the cascade method with the exact same architecture but also simplifies the training process by eliminating the need for two independent trainings and an additional fine-tuning step. Compared to state-of-the-art E2E models, our model achieves slightly lower metrics, but it has a much smaller size and faster inference time. Specifically, LE2E is 90% smaller and $10\times$ faster than JETS while reporting marginally inferior metrics.

Table 3: *Model comparison in terms of memory consumption and computational complexity in a Nvidia A100 GPU*

| Model | Params. | RTF ($\downarrow$) |
|---|---|---|
| VITS [9] | 29.36M | 0.0814 ($\pm$0.0304) |
| JETS [10] | 40.94M | 0.0765 ($\pm$0.0206) |
| LE2E | 3.71M | 0.0084 ($\pm$0.0480) |

## 5. Conclusions and future work

We proposed a lightweight end-to-end text-to-speech (LE2E) architecture that achieves comparable results to VITS and slight worse performance than JETS while being significantly smaller and faster, suitable for on-device applications in low-resource scenarios. Our proposed training paradigm improves existing vocoder architectures and enables the training of a lightweight E2E-TTS system, which replaces the traditional cascade approach and simplifies the training process to a single step. Future research could expand our findings to multi-speaker and/or multi-lingual use-cases, as well as to further explore new discriminator architectures for lightweight TTS models.

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
