# OpenReview forum: "Lightweight End-to-end Text-to-speech Synthesis for low resource on-device applications"
_Interspeech.org/2023/Workshop/SSW — SSW12_

### Official Review · Reviewer_yj1s · 2023-06-04
**Lightweight End-to-end Text-to-speech Synthesis for low resource on-device applications: a novel E2E TTS model based on a joint training of a lightweight acoustic and vocoder model in a single efficient arch. Significantly smaller (90%) than other state of the art E2E solutions (VITS, JETS) while maintaining comparable performance.**

**Rating:** 9
**Confidence:** 3

**Review:**

- Authors propose a new E2E TTS model based on a joint training of a lightweight acoustic and vocoder model in a single efficient arch. Significantly smaller (90%) than other state of the art E2E solutions while maintaining comparable performance. They define a novel joint training paradigm of Lightspeech and multiband MelGAN which outperforms its originally two-step approach while only requiring a single joint training scheme. They also propose an upgraded loss objective based on recent GAN speech discriminator. Objective and Subjective evaluation show that the system achieve MOS comparable with VITS and slightly worse than JETS, while being much more memory efficient and faster.
- The method is novel and definitely relevant for the SSW audience;
- The work is technically correct and sufficient details (training dataset) are provided to allow experiment.
- References are good and up to date and the presentation is really clear with good figures and tables and the results are well described and explained;
- Would be great to have some samples available.

---

### Official Review · Reviewer_Rmhm · 2023-06-04
**Lightweight End-to-end Text-to-speech Synthesis for low resource on-device applications**

**Rating:** 6
**Confidence:** 4

**Review:**

-Key Strength of the paper

Lightweight TTS is a very important issue in neural TTS

The authors propose to combine two lightweight acoustic encoder and decoder for neural TTS, that are optimized jointly by using multiple losses

The proposed model has about 10% of the parameters used by other existing neural TTS such as VITS or JETS, with similar results obtained on LJSpeech

-Main Weakness of the paper

The results that are reported held for single speaker TTS, but what about multi-speaker TTS ?
The expressivity of existing neural TTS (and their complexity) aims at modelling the variability of multiple speakers, multiple styles, etc...
To be convincing, the authors should provide the same experimental comparison with multi-speakers datasets

-Novelty/Originality, taking into account the relevance of the work for the SSW audience

The proposed architecture is the combination of two existing networks with a joint optimisation. The originality is then rather moderate

-Technical Correctness, is the work technically and/or scientifically solid? Are sufficient details provided to allow any experiments to be reproduced or equivalent experiments run?

Yes, mostly

---

### Decision · Program_Chairs · 2023-06-14

**Decision:**

Accept

**Comment:**

SSW2003 received 45 papers. The acceptance rate is 82%. We are pleased to inform you that your paper has been accepted by the SSW2023 Program Committee. Please read the reviews carefully and submit your camera-ready paper by June 28th. Most reviewers performed a detailed review. Please answer to their questions and consider their comments. Note that camera-ready papers are credited with one extra page to allow authors to consider reviewers’ suggestions. So max 7 pages in total including figures & refs.
The deadline for submitting the revised version (with full non-anonymized authors and refs!) is 28th June.